# Critical Limit of Manganese for Soybean in Soils of Three Agro-Ecological Zones of Bangladesh

Harun Or Rashed [1,2], Mohammad Mofizur Rahman Jahangir [1], M. Abul Hashem [1], Jannatul Ferdous [1], M. Abdul Kader [3,4], Zakaria M. Solaiman [5] and Tahsina Sharmin Hoque [1,*]

1 Department of Soil Science, Bangladesh Agricultural University, Mymensingh 2202, Bangladesh; haun.srdi@gmail.com (H.O.R.); mmrjahangir@bau.edu.bd (M.M.R.J.); mahashem@bau.edu.bd (M.A.H.); jannat.ss@bau.edu.bd (J.F.)
2 Soil Resources Development Institute, Divisional Laboratory, Sylhet 3100, Bangladesh
3 Agriculture and Food Technology Discipline, School of Agriculture, Geography, Environment, Ocean and Natural Sciences, University of South Pacific, Apia 1343, Samoa; mdabdul.kader@usp.ac.fj
4 Agriculture Discipline, College of Science, Health, Engineering and Education, Murdoch University, Perth, WA 6009, Australia
5 UWA School of Agriculture and Environment, UWA Institute of Agriculture, The University of Western Australia, Perth, WA 6009, Australia; zakaria.solaiman@uwa.edu.au
* Correspondence: tahsinasharmin@bau.edu.bd

**Abstract:** Manganese (Mn) is an essential micronutrient for plants, which influences multiple physiological systems. Determination of the critical limit (CL) of Mn in the soil is necessary for Mn fertilizer application as this limit delineates the deficiency, optimum, and toxicity ranges of Mn. A pot experiment was performed in the winter season with 20 soils collected from three Agro-Ecological Zones (AEZs) of Bangladesh to determine the CL of Mn for soybean (*Glycine max* L.). Manganese was applied in soil @ 0, 1, 2, and 4 ppm, and the experiment was laid out in a factorial and completely randomized design with three replications. The CL of Mn was determined by the Cate-Nelson graphical and statistical approach and was found 3.60 and 3.55 $\mu g\ g^{-1}$, respectively. Applying 1 $\mu g\ g^{-1}$ Mn in soil significantly enhanced root and shoot weight, as well as seed yield of soybean compared to no Mn application. In field conditions, a significant positive response of soybean yield was found up to the CL of Mn. The findings of the study could help predict possible Mn deficiency in soil and soybean response to Mn fertilizer, which is important in decision-making for efficient fertilizer management practices to ensure the yield potential of soybeans.

**Keywords:** manganese; critical limit; soybean; agro-ecological zones; soil





## 1. Introduction

Soybean (*Glycine max* L. Merrill) is a popular oil seed crop with high food value. It is an excellent source of easily digestible protein, which complements the stable, rich diet in humans [1]. Mature raw soybean seed contains 36.5% protein, 20% lipids, 30% carbohydrates, and 9% dietary fiber [2]. Soybean was grown in 57,646 ha of land in Bangladesh with a production of 91,177 million tons having a yield of 258 kg/ha in the year 2020–2021 [3]. Soybean is a drought [4] and salinity-tolerant crop, thus suitable to cultivate in coastal areas in the Noakhali, Lakshmipur, and Bhola districts of Bangladesh, where cultivation of other crops during the dry winter season is not suitable. New methods and protocols should be described in detail, while well-established methods can be briefly described and appropriately cited. The nodule bacteria (*Bradyrhizobium japonicum*) in the roots of soybean plants can fix atmospheric nitrogen (N), which plays an important role in soil fertility enrichment [5]. An estimated 2.5 to 2.6 million tons of soybean meal are consumed in Bangladesh each year [6]. The country imports an estimated 0.35–0.04 million tons of soybean meal per annum [7]. By increasing soybean production, we can meet the demand

for soybean edible oil and poultry feed. Soybean yields in Bangladesh are significantly lower than anticipated levels. Balanced fertilization can help boost current yield and production levels significantly. Soybean yield can be improved by applying a trace amount of Manganese (Mn), as reported by some researchers [8–10].

In plants, Mn is involved in multiple physiological systems such as photosynthesis, respiration, nitrogen assimilation, nitrogen transformation, etc. [11]. Manganese plays a pivotal role in pollen germination, development and elongation of pollen tube and root cell and in resistance against pathogens [12]. Manganese deficit causes interveinal chlorosis in younger leaves [13] but Mn toxicity shows chlorotic and necrotic spots on older leaves [14] affecting metabolic processes like enzyme activities leading to sterility. Some factors including soil reaction, organic matter content, aeration and moisture status influence Mn availability in soil [15,16]. Manganese availability rises with the fall of soil pH. Manganese deficiency is frequently observed in soils with high pH such as alkaline and calcareous soils and also in poorly aerated soils [17].

For balanced fertilizer application, it is necessary to determine the critical concentration of an essential nutrient element, as this concentration distinguishes deficiency from sufficiency. The critical limit of nutrients is not constant for all the soils and crops and is affected by soil texture, crop cultivar, occasional variation, and soil testing approaches. Therefore, it is necessary to determine the location or crop-specific critical limit for individual nutrients for an area, region, or country. To identify soils of particular areas as deficient and non-deficient of Mn, the determination of the critical limit of Mn in the soil is necessary. This will also help formulate Mn fertilizer recommendations for better crop performance.

Soybean is particularly a Mn-loving crop [18]. In Bangladesh, there is no information concerning Mn fertilizer used on crops. Manganese is used as a fertilizer for soybeans in the Indian subcontinent. Different critical values of available Mn in a particular soil for soybeans are reported for different extractants. Moreover, these limits varied with the soil as well as the extractants used within the same soil [19]. Soil conditions differ from region to region and can vary considerably even within a small field. Therefore, critical deficiency levels determined for one soil may differ from another soil, especially when large geographical areas are considered. In 1997, the critical limit of Mn in Bangladesh soils (Eastern Gangetic Plain) was established at 1.0 ppm by the diethylene triamine-penta acetic acid (DTPA) extraction process, which is still in use irrespective of soil type and crops [20]. The critical Mn level of 3.3 $\mu$g g$^{-1}$ was found for soybeans in neighboring India for similar soils (East Gangetic Plain) [21].

Micronutrient deficiency is considered one of the emerging challenges to food and nutrition security in many countries of the world, including Bangladesh. In recent years, there has been a huge pressure on our soil to produce more food, and therefore, micronutrient deficiencies arise as a consequence of soil fertility depletion, causing a hidden hunger. Accordingly, micronutrient fertilizer application has been increased for fulfilling the micronutrient requirement of crops as well as for maintaining sustainable crop productivity [20]. To ensure global food security, a multifaceted and sustainable approach is needed that will increase crop production as well as promote the resource use efficiency of crops. The effective management of micronutrients in the soil-crop system by soil or foliar application may help mitigate micronutrient deficiency in soils and crops, reduce micronutrient malnutrition in humans and animals, and enhance sustainable crop production and crop quality. As Mn is biochemically involved in plant production, maintaining its availability up to an adequate level is imperative for sustainable food security. Therefore, it is crucial to identify the CL of Mn for specific soils and crops to enable the farmers to increase soybean production and to reduce Mn depletion by application of Mn fertilizers when the Mn content in soils goes below the CL. The critical limit of extractable Mn in the soil can be used to predict the crop yield response to the application of Mn using fertilizer. This also aids in determining the amount of Mn fertilizer required for crop growth as it distinguishes deficiency from sufficiency. For better prediction of Mn deficiency, the CL must be refined or revalidated with reference to the present environment, soil characteristics,

and crop species, which will play a major role in decision-making for balanced fertilization and ensuring better crop yield performance. Considering the above facts, an effort was taken to evaluate the critical limit of Mn for soybeans in soils of different Agro-Ecological Zones (AEZs) for balanced fertilization.

## 2. Materials and Methods

### 2.1. Soil Sampling, Preparation and Analysis

Agroecology incorporates ecological principles and ecosystems with the aim of conserving, protecting, and reinforcing natural ecosystems, as well as sustaining and diversifying the countryside [22]. Based on different landforms, soils, hydrology, and agro-climatic conditions, there are 30 agro-ecological zones in Bangladesh [23]. Among them, Old Himalayan Piedmont Plain (AEZ 1), High Ganges River Floodplain (AEZ 11), and Young Meghna Estuarine Floodplain (AEZ 18) are some important agro-ecological zones where contrasting soils are found with diverse physicochemical characteristics. The evaluation of Mn status, its effect on crop growth, and the critical limit of Mn has not yet been explored recently for soybean cultivation in these regions. Soil sample collection was performed at a depth of 0–15 cm from these three AEZs. The represented areas were Ramgoti, Kamalnagar and Subornochar (AEZ 18), Kustia Sadar (AEZ 11), and Dinajpur Sadar (AEZ 1). The sampling locations are depicted in Figure 1. The general features of the studied soils are listed in Table 1.

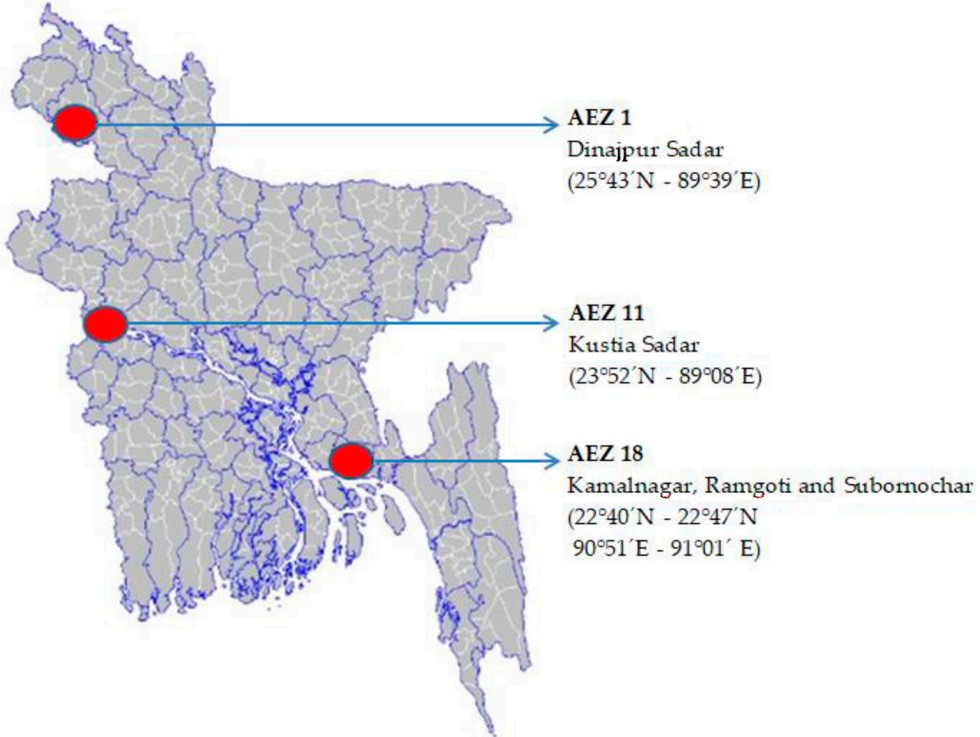

**Figure 1.** Selected areas from three AEZs of Bangladesh with geographical locations.

There were twenty bulk soil samples chosen from a broad range of variables and organized as low to high with respect to the available Mn status based on previous CL [20]. After drying in the shade, the samples were ground by a mortar and pestle, followed by sieving through a 2 mm sieve for analysis. Analyses of the processed soil samples were performed to determine important physicochemical properties such as texture, pH, electrical conductivity (EC) and organic matter (OM) (Table 2). The mechanical analysis of soils was performed using the hydrometer method [24] and the textural classes were identified from Marshall's triangular co-ordinate using USDA system. Glass-electrode pH meter and an electrical conductivity meter were used to measure soil pH and EC,

respectively in 1:2.5 and 1:5 soil-water ratios, respectively [25,26]. Organic matter status of the soil samples was determined using the wet oxidation method [27].

**Table 1.** General information on the sampled soils.

| Soil Samples | Location | Series | AEZ | USDA Soil Taxonomy |
|---|---|---|---|---|
| $S_1$ | Subornochar (22°42′ N 91°01′ E) | Ramgoti | 18 | Typic Haplaquepts |
| $S_2$ | Ramgoti (22°41′ N 90°54′ E) | Nilkamal | 18 | Typic Haplaquepts |
| $S_3$ | Kamalnagar (22°44′ N 90°51′ E) | Ramgoti | 18 | Typic Haplaquepts |
| $S_4$ | Ramgoti (22°43′ N 90°57′ E) | Ramgoti | 18 | Typic Haplaquepts |
| $S_5$ | Kustia Sadar (23°52′ N 89°08′ E) | Ishwardi | 11 | Aeric Haplaquepts |
| $S_6$ | Kamalnagar (22°47′ N 90°52′ E) | Ramgoti | 18 | Typic Haplaquepts |
| $S_7$ | Kustia Sadar (23°52′ N 89°08′ E) | Ishwardi | 11 | Aeric Haplaquepts |
| $S_8$ | Ramgoti (22°43′ N 91°01′ E) | Ramgoti | 18 | Typic Haplaquepts |
| $S_9$ | Kamalnagar (22°44′ N 90°51′ E) | Ramgoti | 18 | Typic Haplaquepts |
| $S_{10}$ | Dinajpur Sadar (25°43′ N 89°39′ E) | Gongachara | 1 | Typic Haplaquepts |
| $S_{11}$ | Dinajpur Sadar (25°43′ N 89°39′ E) | Gongachara | 1 | Typic Haplaquepts |
| $S_{12}$ | Kamalnagar (22°43′ N 90°54′ E) | Ramgoti | 18 | Typic Haplaquepts |
| $S_{13}$ | Kamalnagar (22°43′ N 90°51′ E) | Ramgoti | 18 | Typic Haplaquepts |
| $S_{14}$ | Ramgoti (22°43′ N 90°57′ E) | Ramgoti | 18 | Typic Haplaquepts |
| $S_{15}$ | Kamalnagar (22°45′ N 90°52′ E) | Ramgoti | 18 | Typic Haplaquepts |
| $S_{16}$ | Ramgoti (22°41′ N 90°54′ E) | Nilkamal | 18 | Typic Haplaquepts |
| $S_{17}$ | Ramgoti (22°41′ N 90°54′ E) | Nilkamal | 18 | Typic Haplaquepts |
| $S_{18}$ | Kamalnagar (22°43′ N 90°51′ E) | Ramgoti | 18 | Typic Haplaquepts |
| $S_{19}$ | Dinajpur Sadar (25°43′ N 89°39′ E) | Gongachara | 1 | Typic Haplaquepts |
| $S_{20}$ | Ramgoti (22°40′ N 90°54′ E) | Nilkamal | 18 | Typic Haplaquepts |

**Table 2.** Physical and chemical properties of the experimental soils.

| Soil Samples | pH | EC (dS m$^{-1}$) | OM (%) | Textural Class |
|---|---|---|---|---|
| $S_1$ | 5.37 | 1.01 | 1.72 | Silt loam |
| $S_2$ | 6.56 | 1.06 | 2.02 | Silt loam |
| $S_3$ | 5.76 | 4.01 | 1.97 | Silt loam |
| $S_4$ | 6.53 | 3.50 | 1.72 | Silt loam |
| $S_5$ | 7.79 | 0.27 | 2.32 | Clay loam |
| $S_6$ | 6.68 | 4.15 | 2.24 | Silt loam |
| $S_7$ | 7.85 | 0.23 | 2.81 | Clay loam |
| $S_8$ | 6.52 | 0.93 | 2.24 | Silt loam |
| $S_9$ | 5.89 | 3.86 | 2.45 | Silt loam |
| $S_{10}$ | 5.30 | 0.15 | 1.16 | Loam |
| $S_{11}$ | 5.20 | 0.11 | 0.86 | Sandy loam |
| $S_{12}$ | 6.59 | 1.20 | 2.42 | Silt loam |
| $S_{13}$ | 5.46 | 1.07 | 2.16 | Silt loam |
| $S_{14}$ | 6.61 | 4.08 | 2.24 | Silt loam |
| $S_{15}$ | 6.44 | 3.86 | 2.38 | Silt loam |
| $S_{16}$ | 6.60 | 1.00 | 2.19 | Silt loam |
| $S_{17}$ | 6.77 | 1.32 | 2.53 | Silt loam |
| $S_{18}$ | 5.31 | 1.34 | 2.16 | Silt loam |
| $S_{19}$ | 5.50 | 0.26 | 0.79 | Sandy loam |
| $S_{20}$ | 6.41 | 1.22 | 2.39 | Silt loam |
| Mean | 6.26 | 2.04 | 1.73 | |

The status of both macro and micronutrients in soils is shown in Table 3. Among the macronutrients, total N was measured using micro-Kjeldahl technique [28], available phosphorus (P) using Olsen method (for neutral and alkaline soil) [29] and Bray and Kurtz method (for acid soil) [30], whereas available Sulphur (S) was estimated by extraction with CaCl$_2$ solution (15%) followed by turbidimetric determination using a spectrophotometer [31]. For measurement of the exchangeable bases viz. Potassium (K), Calcium (Ca), and Magnesium (Mg), 1 M NH$_4$OAc solution having pH 7 was used [32]. On the other hand, available soil micronutrients, namely Copper (Cu), Iron (Fe), Mn, and Zinc

(Zn), were assessed by 0.005 M DTPA extraction technique using an atomic absorption spectrophotometer [33]. Available Boron (B) was measured using a hot water 0.02 M $CaCl_2$ solution (1:2) extraction followed by determination using azomethine-H [34].

**Table 3.** Nutrient status of soils collected from different AEZs.

| Soil Samples | N % | K | Ca cmol (+) $Kg^{-1}$ | Mg | P | B | S | Cu $\mu g\ g^{-1}$ | Fe | Mn | Zn |
|---|---|---|---|---|---|---|---|---|---|---|---|
| $S_1$ | 0.09 | 0.12 | 4.11 | 1.97 | 7.40 | 0.12 | 55.7 | 1.34 | 55.6 | 1.78 | 0.88 |
| $S_2$ | 0.10 | 0.16 | 4.03 | 2.29 | 11.0 | 0.10 | 31.8 | 1.84 | 59.9 | 2.61 | 0.59 |
| $S_3$ | 0.10 | 0.24 | 4.62 | 2.57 | 6.90 | 0.11 | 35.9 | 2.06 | 87.1 | 2.72 | 0.37 |
| $S_4$ | 0.09 | 0.12 | 4.11 | 2.03 | 1.20 | 0.18 | 76.1 | 0.91 | 32.6 | 2.74 | 0.32 |
| $S_5$ | 0.12 | 0.52 | 16.0 | 3.53 | 6.02 | 0.65 | 23.1 | 5.29 | 71.1 | 2.81 | 1.05 |
| $S_6$ | 0.11 | 0.23 | 4.02 | 3.36 | 1.80 | 0.15 | 74.5 | 1.95 | 33.0 | 2.88 | 0.42 |
| $S_7$ | 0.14 | 0.57 | 15.3 | 3.28 | 5.53 | 0.60 | 28.1 | 5.56 | 76.5 | 3.07 | 1.39 |
| $S_8$ | 0.11 | 0.16 | 4.76 | 2.74 | 7.50 | 0.14 | 29.9 | 1.58 | 57.2 | 3.14 | 0.66 |
| $S_9$ | 0.12 | 0.22 | 4.37 | 2.61 | 8.20 | 0.09 | 27.8 | 2.55 | 135 | 3.19 | 0.63 |
| $S_{10}$ | 0.06 | 0.14 | 1.94 | 0.56 | 5.50 | 0.20 | 10.0 | 1.53 | 101 | 3.23 | 1.40 |
| $S_{11}$ | 0.04 | 0.11 | 1.15 | 0.36 | 7.60 | 0.24 | 12.1 | 0.78 | 57.6 | 3.86 | 0.60 |
| $S_{12}$ | 0.12 | 0.20 | 2.15 | 1.17 | 1.80 | 0.18 | 51.1 | 2.16 | 120 | 4.08 | 0.68 |
| $S_{13}$ | 0.11 | 0.21 | 2.51 | 1.24 | 2.00 | 0.15 | 49.0 | 2.34 | 118 | 4.11 | 0.70 |
| $S_{14}$ | 0.11 | 0.13 | 4.30 | 2.30 | 2.00 | 0.21 | 84.5 | 1.19 | 30.6 | 4.12 | 0.33 |
| $S_{15}$ | 0.12 | 0.20 | 4.01 | 3.23 | 2.20 | 0.16 | 55.3 | 2.65 | 35.7 | 4.13 | 0.62 |
| $S_{16}$ | 0.11 | 0.12 | 4.56 | 2.57 | 5.80 | 0.09 | 51.9 | 1.51 | 47.0 | 4.46 | 0.98 |
| $S_{17}$ | 0.13 | 0.12 | 5.95 | 2.52 | 3.20 | 0.16 | 74.6 | 1.43 | 32.2 | 5.33 | 1.22 |
| $S_{18}$ | 0.11 | 0.23 | 2.83 | 1.45 | 3.70 | 0.17 | 38.0 | 2.63 | 129 | 5.36 | 0.77 |
| $S_{19}$ | 0.04 | 0.27 | 2.48 | 1.16 | 3.50 | 0.15 | 17.9 | 1.28 | 53.2 | 5.99 | 1.70 |
| $S_{20}$ | 0.12 | 0.12 | 6.66 | 2.78 | 3.20 | 0.18 | 66.5 | 1.75 | 35.0 | 6.76 | 1.48 |
| Mean | 0.10 | 0.21 | 4.99 | 2.12 | 4.80 | 0.20 | 44.7 | 2.12 | 68.3 | 3.82 | 0.84 |

*2.2. Pot Trial*

In Rabi season of 2019–2020, a pot trial was conducted at the Soil Science Field Laboratory of Bangladesh Agricultural University, Mymensingh. The experiment was conducted with 20 soils and four levels of Mn ($Mn_0$, $Mn_1$, $Mn_2$, and $Mn_4$) viz. 0, 1, 2, and 4 $\mu g\ g^{-1}$ in a factorial completely randomized design (CRD) with three replications. A number of 240 plastic pots with an eight-kilogram capacity (24 cm in diameter) were filled with 20 soils collected from different locations (mentioned in Table 1) of AEZ 1, 11 and 18, where soil moisture content was maintained at field capacity (@ 25% moisture *v/v*). According to FRG [20], N, P, K, S, Zn and B were applied from urea, triple super phosphate, muriate of potash, gypsum, zinc sulphate and boric acid, respectively at the dose of 27, 30, 40, 14, 1 and 0.5 kg $ha^{-1}$, respectively. All these chemical fertilizers were applied at full doses to all the soils. Manganese was applied in soils from analytical grade Mn sulphate ($MnSO_4.H_2O$). To add 0, 1, 2 and 4 $\mu g$ Mn $g^{-1}$ soil, the required amount of $MnSO_4.H_2O$ was 0 $\mu g\ g^{-1}$ soil (i.e., 0 kg $ha^{-1}$), 3.07 $\mu g\ g^{-1}$ soil (i.e., 6.14 kg $ha^{-1}$), 6.14 $\mu g\ g^{-1}$ soil (i.e., 12.28 kg $ha^{-1}$), and 12.28 $\mu g\ g^{-1}$ soil (i.e., 24.56 kg $ha^{-1}$). BARI Soybean-6 was used as a test crop. Six seeds were sown in each pot and after emergence three seedlings were allowed to grow. Intercultural operations such as weeding, irrigation and pest control were performed as per requirement of the crop. The soybean plants were harvested after 116 days of sowing. The harvested root, shoot and seed samples were prepared and dried at 65 °C in an oven till constant weight was obtained. The dry weight of root, shoot and seed of soybean for each pot was recorded. The seed yield was calculated at 14% moisture basis.

*2.3. Determination of Critical Limit of Mn Using the Graphical Method*

The CL of Mn in the studied soils was determined by the graphical method, as suggested by Cate and Nelson [35]. The yield of soybean was obtained at the 100%

flowering stage of the crop and was converted into Bray's percent yield by using the following equation.

$$\text{Bray's \%yield} = \frac{\text{Yield without Mn}}{\text{Yield at optimum Mn treatment}} \times 100$$

The critical level of nutrients in soil was derived by plotting the nutrient on 'X' axis and Bray's percent yield on 'Y' axis as described by Prasad et al. [36]. A cross is placed over the data and moved to the upper left and lower right to have a minimum number of points, as suggested by Cate and Nelson [35]. In this method, the dividing line between two groups was drawn roughly, and the lines for vertical and horizontal directions on a scatter diagram were superimposed subjectively. In order to solve it, a statistical model was followed, which was developed by Cate and Nelson [37] in accordance with Waugh's recommendations [38].

### 2.4. Determination of Critical Limit of Mn Using the Statistical Method

The CL of Mn in the soils under study was also calculated using a statistical approach as developed by Cate and Nelson [37], where the initial soil test values of Mn were in ascending order, and Bray's percent yield was placed against each soil test Mn value. The pairs of soil test values and relative yield were kept in order throughout the analyses [39,40]. According to Siva Prasad et al. [36], the correction factor (C.F.) and a total corrected sum of squares (T.C.S.S.) were calculated from Bray's percent yield by using the following formula

$$\text{C.F.} = \frac{\{\sum Y\}^2}{n} = \frac{\sum\{Y_1+Y_2+Y_3\ldots\ldots+Y_n\}}{n} \text{ and}$$

$$\text{T.C.S.S.} = \sum_{i=1}^{n} Y^2 - \text{C.F.} = \sum\left(Y_1^2 + Y_2^2 + Y_3^2 \ldots\ldots + Y_n^2\right) - \text{C.F.}$$

The data were grouped into two classes, i.e., the samples S1 to S10 were considered population 1 and S11 to S20 were considered population 2. Plant available Mn was included in population 1 = $\frac{P_1+p_2+\ldots\ldots\ldots P_n}{P}$.

The corrected sum of squares of derivation from the mean of population 1 (CSSI) = $\sum P_1^2 + P_2^2 + \ldots\ldots\ldots P_n^2 - \frac{\sum(P_1+\ldots\ldots\ldots P_n)}{n}$.

If Kn is the number of observations in population 2, then the mean relative yield in population 2 = $\frac{K_1+k_2\ldots\ldots\ldots+K_n}{n}$.

The PCL, i.e., postulated critical limit (split between two populations) was calculated as follows

$$\text{PCL} = \frac{\text{Least value in population 1} - \text{1st value in population 2}}{2}$$

As a consequence, the predictability value $R^2$ was computed from the following equation.

$R^2$ = {TCSS − (CSS1 + CSS2)}/TCSS where
TCSS = Total Corrected Sum of Squares
CSS1 = Corrected Sum of Squares of Population 1
CSS2 = Corrected Sum of Squares of Population 2

From the highest $R^2$, the postulated critical limit was found.

### 2.5. Validation Study at Field Condition

A field trial was conducted to assess soybean (BARI Soybean-6) response to Mn fertilization at AEZ 18 in Subornochar upazila (22°42′ N, 91°01′ E) during the Rabi season of 2020–2021 where available soil Mn was 1.78 mg kg$^{-1}$ (S$_1$). For the field study, the same fertilizers were used, as mentioned in Section 2.2. Manganese was applied from MnSO$_4$ on the basis of soil test values considering CL. The treatments were 50% CL (1.8 µg Mn g$^{-1}$

soil, i.e., 3.6 kg Mn ha$^{-1}$), 75% CL (2.7 µg Mn g$^{-1}$ soil, i.e., 5.4 kg Mn ha$^{-1}$), 100% CL (3.6 µg Mn g$^{-1}$ soil, i.e., 7.2 kg Mn ha$^{-1}$) and 125% CL (4.5 µg Mn g$^{-1}$ soil, i.e., 8.9 kg Mn ha$^{-1}$). Each treatment was replicated four times in a randomized complete block design (RCBD). During final land preparation, a basal dose of 27 kg ha$^{-1}$ of N, 30 kg ha$^{-1}$ of P, 40 kg ha$^{-1}$ of K, 14 kg ha$^{-1}$ S, 1 kg ha$^{-1}$ of Zn and 0.5 kg ha$^{-1}$ of B were applied according to the soil test basis from urea, TSP, MoP, gypsum, zinc sulphate and boric acid, respectively. The unit plot size was 4 m × 3 m and the total number of the unit plots was 16 (4 × 4). Manganese was applied as $MnSO_4 \cdot H_2O$ by mixing with seeds at sowing. The required amount of $MnSO_4 \cdot H_2O$ was 5.53 µg g$^{-1}$ soil (i.e., 11.06 kg ha$^{-1}$) for 50% CL, 8.29 µg g$^{-1}$ soil (i.e., 16.58 kg ha$^{-1}$) for 75% CL, 11.05 µg g$^{-1}$ soil (i.e., 22.10 kg ha$^{-1}$) for 100% CL, and 13.8 µg soil (i.e., 27.60 kg ha$^{-1}$) for 125% CL. Maintaining a distance of 4 cm, the seed sowing was performed @ 55 kg ha$^{-1}$. For proper development of the crop, suitable intercultural and management operations were conducted. At the full matured stage, harvesting was performed after 116 DAS. The seed yield was calculated considering 14% moisture level and the yield data were analyzed in the same manner as the pot experiment.

### 2.6. Analysis of Data

Statistical analysis of the data was carried out by F-test to evaluate the significance of the treatment effects [41]. Using DMRT (Duncan's Multiple Range Test), the comparison of the treatment means was performed. With the help of a computer package program, "MSTAT-C", the analysis of variance (ANOVA) for different parameters was performed.

## 3. Results

### 3.1. General Characteristics of Soils Used in the Study

Four major characteristics of soil, viz. pH, EC, organic matter content, and textural classes, are mentioned in Table 2. Soil pH differed from 5.20 to 7.85 (mean value 6.26), EC varied from 0.11–4.15 dS m$^{-1}$ (mean value 1.73 dS m$^{-1}$), and organic matter status ranged from 0.79 to 2.81% (mean value 2.04%). Based on pH, the soils of the studied locations were strongly acidic to slightly alkaline in reaction. Dinajpur Sadar (AEZ 1) recorded the lowest pH of 5.20, whereas the highest pH (7.85) was observed in Kustia Sadar (AEZ 11). The EC classes were non-saline to slightly saline. The maximum EC value (4.15 dS m$^{-1}$) was found in Kamalnagar upazila under AEZ 18, whereas the minimum EC (0.11 dS m$^{-1}$) was recorded in Dinajpur Sadar of AEZ 1. About 85% of the soils were medium, 5% of soils were low, and the rest 10% soils were very low in organic matter status. Again, the organic matter content in soil was minimum (0.79%) in Dinajpur Sadar, whereas the maximum content (2.81%) was observed in Kustia Sadar. Most of the soils were silt loam in texture, although sandy loam and loam soils were found in Dinajpur Sadar and clay loam soils were recorded in Kustia Sadar.

The status of both macro and micronutrients in the selected soils is shown in Table 3. In the soil set, total N content varied from 0.04 to 0.14%, whereas exchangeable K, Ca, and Mg status ranged from 0.11 to 0.57, 1.15 to 16.0, and 0.36 to 0.53 cmol (+) kg$^{-1}$, respectively. Available P, B, and S amounts differed from 1.20 to 11.0, 0.09 to 0.65, and 10.0 to 84.5 µg g$^{-1}$, respectively. The status of available Zn, Cu, Fe, and Mn of the soils fluctuated from 0.32 to 1.70, 0.78 to 5.56, 30.6 to 135, and 1.78 to 6.76 µg g$^{-1}$, respectively, in the studied soil sets having mean values of 0.84, 2.12, 68.4 and 3.82 µg g$^{-1}$, respectively. The Zn level was very low to optimum. Considering the critical value of Mn in soils (1.0 µg g$^{-1}$), 5% of soils ($S_1$) was grouped into medium (1.51–2.25 µg g$^{-1}$ Mn), 25% ($S_2$–$S_6$) optimum (2.26–3.0 µg g$^{-1}$ Mn), 20% ($S_7$–$S_{10}$) high (3.10–3.75 µg g$^{-1}$ Mn) and 50% ($S_{11}$–$S_{20}$) very high classes (>3.75 µg g$^{-1}$ Mn) according to FRG [20]. On the other hand, the status of both Cu and Fe was very high in soils of the selected AEZs.

### 3.2. Effect of Mn on Root, Shoot, and Seed Weight of Soybean

The data mentioned in Table 4 suggest that various levels of Mn applied to different soils had a significant impact on soybean root, shoot, and seed weight. Manganese influenced the root weight of soybeans. The highest root weight of 2.21 g pot$^{-1}$ and the lowest root weight of 1.89 g pot$^{-1}$ were achieved from $Mn_1$ and $Mn_0$ treatments, respectively. The difference between these treatments was significant. The present study revealed that Mn possesses a considerable effect on the shoot weight of soybeans grown in the soils of selected AEZs. The significantly highest shoot weight (16.6 g pot$^{-1}$) was obtained in $Mn_1$-treated soils over control, which was followed by $Mn_2$ and $Mn_4$. As the dose of Mn increased, the shoot weight gradually decreased. The effect of Mn was considerable in increasing the seed weight of soybeans. The $Mn_1$ treatment produced the highest seed weight of 11.8 g pot$^{-1}$ followed by $Mn_2$ and $Mn_4$ over control.

**Table 4.** Root, shoot, and seed weight of soybeans are affected by different levels of Mn in different AEZs.

| Treatments | Root | Shoot | Seed |
|---|---|---|---|
| | | Dry Weight (g pot$^{-1}$) | |
| AEZs | | | |
| AEZ 18 | 2.03 | 15.9 | 11.3 |
| AEZ 11 | 2.08 | 15.9 | 11.3 |
| AEZ 1 | 2.05 | 15.7 | 11.0 |
| LSD | 0.17 | 0.49 | 0.66 |
| SE ($\pm$) | 0.09 | 0.96 | 0.34 |
| Mn level | | | |
| $Mn_0$ | 1.89b | 14.2b | 10.0b |
| $Mn_1$ | 2.21a | 16.6a | 11.8a |
| $Mn_2$ | 2.08ab | 16.5a | 11.6a |
| $Mn_4$ | 2.02ab | 15.9a | 11.5a |
| LSD | 0.20 | 1.11 | 0.76 |
| SE ($\pm$) | 0.10 | 0.56 | 0.39 |
| Level of significance | | | |
| AEZ | ns | ns | ns |
| Mn level | ** | *** | *** |
| AEZ $\times$ Mn level | ns | ns | ns |

ns = Not-significant; ** = Significant at 1% level of probability; *** = Significant at 0.1% level of probability. Figures in a column having common letter(s) do not differ significantly according to DMRT.

### 3.3. Critical Limit of Soil Mn for Soybean

Figure 2 shows the graphical representation of relative yield and soil test available Mn in the form of XY scattered graph points. A horizontal line was drawn at that point by averaging Bray's percentage yield. A perpendicular line was also drawn, and the point where the perpendicular line crosses the X axis is the soil critical level. Thus, according to the graphical method, the scatter diagram indicated the critical limit as 3.60 µg g$^{-1}$, below which the response of Mn application to soil may be expected in the case of soybean. The Bray's yield corresponded well with the Mn content of the soil, and it varied from 65.2 to 95.6% (Table 5).

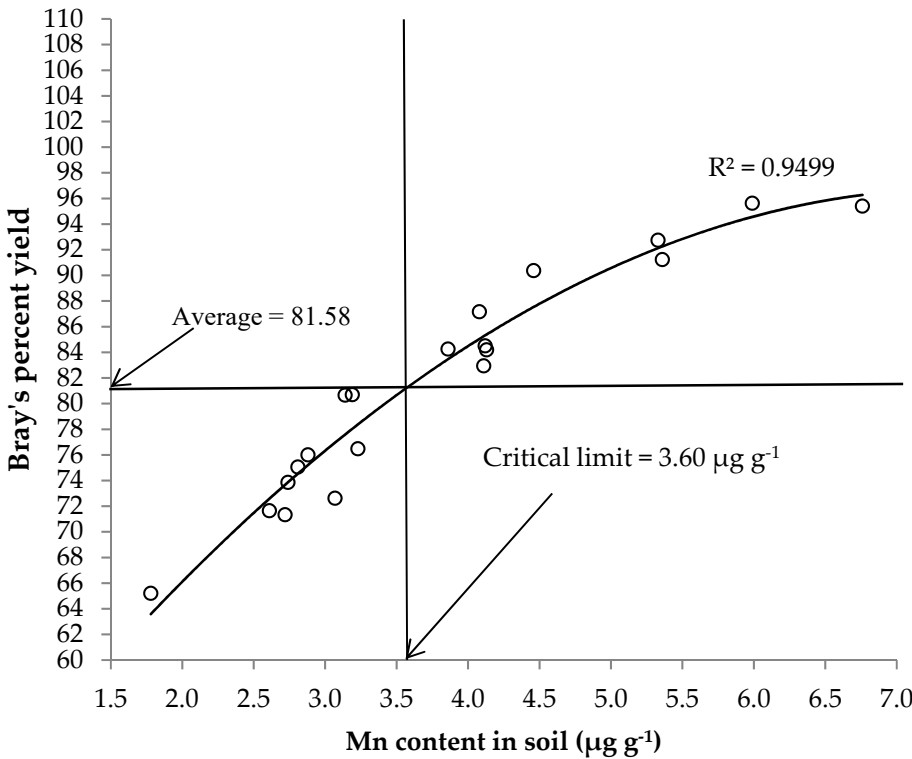

**Figure 2.** Critical limit of Mn for soybean production in soils of different AEZs.

**Table 5.** Initial soil Mn, Seed yield (g pot$^{-1}$) with and without Mn, and Bray's %yield.

| Initial Soil Mn ($\mu g\ g^{-1}$ soil) | Mn Levels ($\mu g\ g^{-1}$ Soil) | | | | Bray's %Yield |
| --- | --- | --- | --- | --- | --- |
| | $Mn_0$ | $Mn_1$ | $Mn_2$ | $Mn_4$ | |
| 1.78 | 9.14 | 14.0 | 13.3 | 11.1 | 65.2 |
| 2.61 | 9.71 | 11.7 | 12.9 | 13.6 | 71.6 |
| 2.72 | 9.11 | 12.8 | 9.62 | 10.6 | 71.3 |
| 2.74 | 9.02 | 12.2 | 10.6 | 9.06 | 73.8 |
| 2.81 | 9.29 | 11.7 | 11.9 | 12.4 | 75.0 |
| 2.88 | 9.04 | 10.2 | 11.9 | 9.60 | 76.0 |
| 3.07 | 9.29 | 11.6 | 11.6 | 12.8 | 72.6 |
| 3.14 | 10.6 | 12.8 | 13.2 | 13.1 | 80.6 |
| 3.19 | 9.02 | 10.4 | 11.2 | 9.33 | 80.7 |
| 3.23 | 9.75 | 12.8 | 11.2 | 11.7 | 76.5 |
| 3.86 | 9.16 | 9.74 | 10.9 | 9.45 | 84.2 |
| 4.08 | 9.65 | 9.71 | 10.6 | 11.1 | 87.2 |
| 4.11 | 9.20 | 10.1 | 11.1 | 9.80 | 82.9 |
| 4.12 | 9.35 | 10.1 | 11.1 | 9.39 | 84.5 |
| 4.13 | 11.3 | 12.0 | 13.5 | 10.4 | 84.2 |
| 4.46 | 12.0 | 13.2 | 12.1 | 13.3 | 90.4 |
| 5.33 | 11.9 | 12.8 | 12.2 | 12.5 | 92.7 |
| 5.36 | 12.4 | 13.6 | 12.9 | 13.3 | 91.2 |
| 5.99 | 12.4 | 13.0 | 11.4 | 10.9 | 95.6 |
| 6.76 | 11.8 | 12.2 | 12.4 | 12.1 | 95.4 |

By statistical method, the critical limit of available Mn in soil for soybean was found to be 3.55 $\mu g\ g^{-1}$ (Table 6). The soil available Mn content varied from 1.78 to 6.76 $\mu g\ g^{-1}$, and Bray's percent yield varied from 65.19 to 95.61, and these were used to fix the critical limit of Mn in soil. The $R^2$ values ranged from 0.23 to 0.73. The highest $R^2$ (0.73) showed the postulated critical limit of Mn for soybean, and it was 3.55 $\mu g\ g^{-1}$ soil.

**Table 6.** Statistical method showing the critical limit of Mn in soil for soybean.

| Soil Samples | Soil Available Mn | Bray's %Yield | Last Value of Soil Mn in P1 | Mean Bray's %Yield in P1 | CSS1 | Mean Bray's %Yield in P2 | CSS2 | PCL | $R^2$ |
|---|---|---|---|---|---|---|---|---|---|
| $S_1$ | 1.78 | 65.19 | | | | | | | |
| $S_2$ | 2.61 | 71.61 | 2.61 | 68.40 | 20.61 | 2437.55 | 1209.86 | 2.67 | 0.23 |
| $S_3$ | 2.72 | 71.34 | 2.72 | 71.48 | 26.37 | 2572.78 | 1075.05 | 2.73 | 0.31 |
| $S_4$ | 2.74 | 73.81 | 2.74 | 72.58 | 41.09 | 2724.16 | 979.61 | 2.78 | 0.36 |
| $S_5$ | 2.81 | 75.04 | 2.81 | 74.43 | 57.67 | 2894.81 | 896.25 | 2.85 | 0.40 |
| $S_6$ | 2.88 | 68.33 | 2.88 | 71.69 | 65.51 | 3088.93 | 617.17 | 2.98 | 0.57 |
| $S_7$ | 3.07 | 72.58 | 3.07 | 70.46 | 67.97 | 3311.39 | 433.99 | 3.11 | 0.69 |
| $S_8$ | 3.14 | 80.66 | 3.14 | 76.62 | 147.46 | 3568.59 | 395.43 | 3.17 | 0.66 |
| $S_9$ | 3.19 | 80.68 | 3.19 | 80.67 | 209.59 | 3869.77 | 350.16 | 3.21 | 0.65 |
| $S_{10}$ | 3.23 | 76.47 | 3.23 | 78.58 | 218.93 | 4227.62 | 211.23 | **3.55** | 0.73 |
| $S_{11}$ | 3.86 | 84.27 | 3.86 | 80.37 | 322.99 | 4659.14 | 188.10 | 3.97 | 0.68 |
| $S_{12}$ | 4.08 | 87.12 | 4.08 | 85.70 | 467.97 | 5189.93 | 182.57 | 4.10 | 0.59 |
| $S_{13}$ | 4.11 | 82.96 | 4.11 | 85.04 | 518.09 | 5859.40 | 131.93 | 4.12 | 0.59 |
| $S_{14}$ | 4.12 | 84.46 | 4.12 | 83.71 | 582.08 | 6729.70 | 88.42 | 4.13 | 0.58 |
| $S_{15}$ | 4.13 | 84.18 | 4.13 | 84.32 | 633.59 | 7907.13 | 22.62 | 4.30 | 0.59 |
| $S_{16}$ | 4.46 | 90.38 | 4.46 | 87.28 | 795.29 | 9588.51 | 13.60 | 4.90 | 0.49 |
| $S_{17}$ | 5.33 | 92.76 | 5.33 | 91.57 | 998.46 | 12,186.3 | 12.32 | 5.35 | 0.37 |
| $S_{18}$ | 5.36 | 91.2 | 5.36 | 91.98 | 1140.61 | 16,732.6 | 0.03 | 5.68 | 0.28 |
| $S_{19}$ | 5.99 | 95.61 | 5.99 | 93.41 | 1383.03 | 26,733.1 | | 6.38 | |
| $S_{20}$ | 6.76 | 95.38 | 6.76 | 95.50 | 1594.65 | 66,734.1 | | | |

PCL = Postulated critical limit; CSS1 = Corrected sum of squares of derivation from population 1; CSS2 = Corrected sum of squares of derivation from population 2; P1 = Population 1; P2 = Population 2.

*3.4. Yield of Soybean in Case of Field Trial*

Manganese application significantly influenced soybean seed yield, as shown in Figure 3. The maximum yield was 1.96 t ha$^{-1}$ obtained from 100% CL, whereas the minimum yield was 1.65 t ha$^{-1}$ recorded in 50% CL.

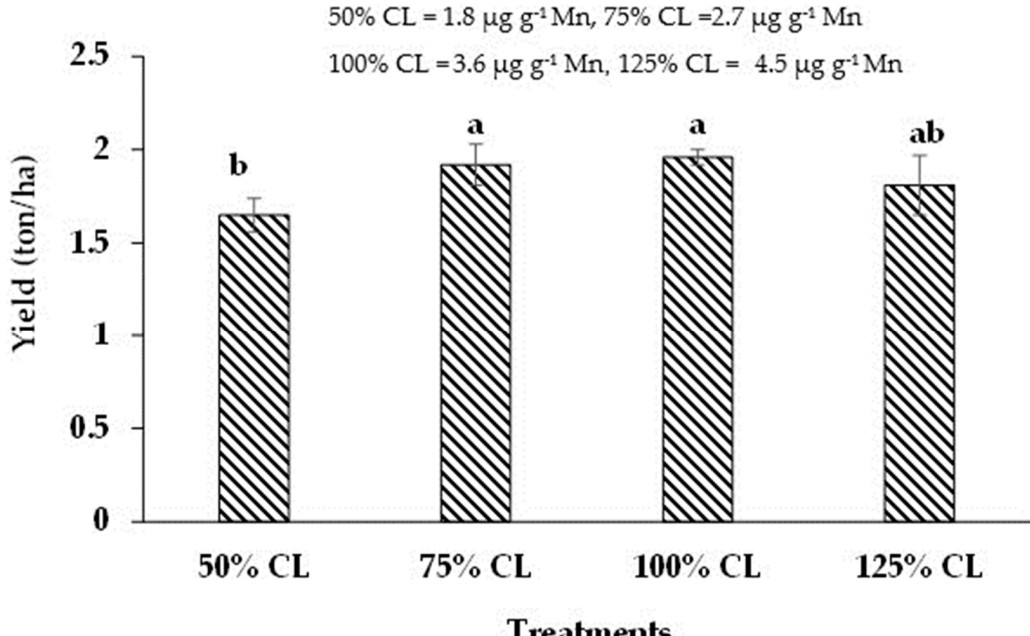

**Figure 3.** Effect of Mn application on soybean seed yield in AEZ 18. Different lowercase letter(s) represent significant difference among treatments according to DMRT.

## 4. Discussion

Twenty typical soils were chosen from 240 soils in different locations, covering three AEZs (AEZ 18, AEZ 11, and AEZ 1) with varying physicochemical properties. The critical limit of soil Mn for soybeans in three selected AEZs was detected in this study. The critical Mn level is a soil test level that indicates the division between responsive and non-responsive conditions to Mn application. According to Siva Prasad [36], this level produces the best separation between soils that give a yield response for crops from those that do not. The critical level is successful in identifying the deficient and sufficient nutrient concentrations for a variety of plant and tree species [42]. Determination of the critical limits of nutrients helps in the validation and interpretation of soil test results and in making precise fertilizer recommendations by soil categorization [43]. The critical levels can be used to estimate the probability of fertilization responses [44]. The different critical values of Mn for soybeans could be attributed to the diversified properties of soils. A variation in sample sites can cause a significant change in nutrient-critical levels, as suggested by Rahman et al. [45]. In the graphical technique, the plotting of calibration data is done as Bray's %yield versus soil test value. In Figure 2, depicting the critical limit of Mn in soil using the graphical method, the points in the upper left quadrangle reflect soil test values that are below the critical limit, indicating that plant response is likely when Mn is added. The points in the lower right quadrant indicate a situation when Mn content exceeds the CL. The horizontal and perpendicular lines were drawn while all points were distributed in the lower left and upper right quadrangle in Figure 2. The perpendicular line crosses the *X*-axis at a value of 3.60, which is considered the CL for available Mn in soils of the three AEZs. Thus, soils having <3.60 $\mu$g Mn g$^{-1}$ soil (i.e., 7.2 kg Mn ha$^{-1}$) will respond to Mn fertilizer application at the flowering stage to achieve yield in the case of soybean.

According to FRG [20], the CL for Mn in soils of different AEZs of Bangladesh was 1 $\mu$g g$^{-1}$, although no record was found for soybean crops. This increase in CL for Mn may be due to adequate Mn status in soil and the interaction of other elements, especially Fe. The critical Mn level for soils used for soybean production in India was reported as 3.3 $\mu$g g$^{-1}$ [21]. The statistical method for determining critical levels of Mn in soil was used to obtain the soil test value received from the soybean crop experiment (Table 6). Isolation of the threshold value of available Mn in the soil can be performed by taking into account the maximum $R^2$ value with the corresponding presumed CL for soybean, which can be pointed out at 3.55 $\mu$g g$^{-1}$. According to some previous reports, the graphical method for estimating CL gives a higher value compared to the statistical method [46]. Either of the methods (graphical or statistical) can be utilized to ascertain the CL of nutrients in the soil, as the values of CL determined using two different methods are very close. To ensure higher crop output, the authors suggest using the maximum critical value calculated from both the graphical and statistical methods, as a significant percentage of soils will not be able to comply with the CL. As reported by Rahman et al. [45], a compatible approach can predict the amount of plant-available nutrients and crops' response to fertilizer application in various types of soils. Therefore, in order to find out the optimum fertilizer demand for a crop, it is crucial to determine CL using two different methodologies.

Manganese is associated with numerous metabolic functions in plant cellular compartments and is of paramount importance as a micronutrient for plant growth. Soybean is a high Mn-responsive crop [47]. In many regions of the world, soybeans are more often deficient in Mn than in other micronutrients and respond well to Mn fertilizers when deficient. Diagnosis and prevention of any nutrient deficiency and toxicity require knowledge of the critical value of that particular element in soil and plant, as suggested by Reddy et al. [42]. The critical limit can be used to find Mn-deficient soils from non-deficient and can provide information on the Mn status of soils. Elucidation of a critical level of Mn delineates the deficiency, optimum, and toxic range of Mn, which is very important with respect to fertilizer application at peak demand and improves the yield of soybeans in Mn-deficient soils. According to Randall et al. [48], soil or foliar application of Mn is most

effective in soybeans when applied at the early blossom or early pod set stage or in multiple applications at these stages.

In this study, soybean root, shoot, and seed weight increased markedly and significantly over control due to Mn addition in soil. Application of Mn at 1 $\mu$g g$^{-1}$ rate gave the highest seed yield and showed the maximum increases in dry weight of different organs of soybean. Further, the Mn application improved the number of pods and seeds in soybeans). The improvement in soybean yield might be due to the role of Mn in enhancing photosynthesis efficiency and carbohydrate (starch) synthesis, as suggested by Mousavi et al. [49]. Ghasemian et al. [8] showed that Mn addition gave the highest seed number and seed weight per plant, pod number, and seed yield, as well as producing the highest biological yield of soybean. A similar effect was found by Kwano et al. [50]. Several workers reported that Mn increased the root, shoot, and yield of different crops. A 25% reduction in photosynthesis and a 20% decrease in dry matter yield in Mn-deficient spinach plants were demonstrated by Bottrill et al. [51]. Arya and Roy [52] found similar results in broad beans. Akay and Uzun [53] reported that fresh yield of broccoli responded positively to Mn addition. In our field study, Mn fertilization significantly increased soybean yields at AEZ 18, although maximum yields were achieved with the margin of critical limit (100% CL) of Mn. The field validation showed that the estimated CL of Mn (3.60 $\mu$g g$^{-1}$) represents the better response of soybean yield.

There is an optimum concentration level at which each nutrient is neither deficient, wasteful, nor toxic. A very low concentration of a micronutrient like Mn can sometimes cause deficiency, and on the contrary, a higher concentration can block the uptake and utilization of other nutrients and can lead to toxicity in the soil and plant. The best strategy to overcome nutrient deficiencies and avoid toxicities is to adopt balanced fertigation and apply fertilizers on the basis of soil tests.

## 5. Conclusions

The critical limit of a nutrient element plays an important role in farm-level decision-making making, especially for balanced fertilization to increase the yield and quality of crops. The present study showed that soil Mn content and Mn application rate had a significant impact on the root, shoot, and seed weight of soybeans. According to the findings, approximately 50% of the soybean-grown soils in three AEZs showed a significant response to added Mn, where Mn deficiency was nil. We conclude that the additional Mn supply would increase soybean yield in soils with an Mn content of less than 3.60 $\mu$g g$^{-1}$. The critical limit for soil available Mn was 3.60 $\mu$g g$^{-1}$, as noted by the graphical method, and 3.55 $\mu$g g$^{-1}$ by statistical method for the selected AEZs. The CL of Mn for soybean showed in the present study was higher in both methods than the previous CL, i.e., 1.0 $\mu$g g$^{-1}$ soil as suggested by the Fertilizer Recommendation Guide. The developed CL from the present study may be used for prediction of Mn deficiency in soils and yield response of soybean to Mn addition which is necessary at farm level planning particularly for the application of balanced nutrient to ensure better soybean production. Nevertheless, it is tough to fix a constant value of critical limit of soil available Mn in various types of soils owing to soil heterogeneity, crop species and different varieties of a given species.

**Author Contributions:** Conceptualization and methodology, T.S.H., M.M.R.J. and M.A.H.; research and data collection, H.O.R.; data analysis, H.O.R. and J.F.; writing—original draft, H.O.R.; writing—reviewing and editing, T.S.H., M.M.R.J., J.F., M.A.H., M.A.K. and Z.M.S.; supervision, T.S.H. and M.M.R.J. All authors have read and agreed to the published version of the manuscript.

**Funding:** The current research was financed by the NATP-2 National Agricultural Technology Program Phase-2 (Project ID: P166344) executed by the Bangladesh Agricultural Research Council (BARC), Dhaka, Bangladesh.

**Institutional Review Board Statement:** Not applicable.

**Informed Consent Statement:** Not applicable.

**Data Availability Statement:** The data are contained within the article.

**Acknowledgments:** The authors gratefully acknowledge the financial support from theWorld Bank and the logistic support from the Department of Soil Science, Bangladesh Agricultural University, to carry out this research work.

**Conflicts of Interest:** The authors declare no conflict of interest.

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
