# Peer review of "Critical Limit of Manganese for Soybean in Soils of Three Agro-Ecological Zones of Bangladesh"

_sustainability, doi:10.3390/su152316410_

Round 1

Reviewer 1 Report

Comments and Suggestions for Authors

This paper on soya beans and agroecological practices is certainly interesting and fill a gap in the literature. 
there are the following issues that can however be improved:

- as it stands, the abstract seems to technical; please rephrase it to make it clearer and more accessible to a general public

- please justify the choice of the case study. This can be done in the methodology section.

- in the same section, please better justify the choice of these methods of data collection and analysis -— 

- finally, please read and discuss in your paper the newly published piece of Ariane Goetz et al on polycentricity governance and agroecology in the Mena region published in international journal of water resources development. I am sure this would be a good addition to your work.

Author Response

Comment:

This paper on soya beans and agroecological practices is certainly interesting and fill a gap in the literature. 

Response:

The authors are thankful to the reviewer for the encouraging comment.

Comment:

- as it stands, the abstract seems to technical; please rephrase it to make it clearer and more accessible to a general public

Response:

According to the reviewer’s comments, the abstract has been revised. Please see Page 1 line 20-31.

Comment:

- please justify the choice of the case study. This can be done in the methodology section.

Response:

According to the reviewer’s comment, the choice of the case study has been justified in the Methodology section. Please see Page 3 line 99-108. 

Comment:

- in the same section, please better justify the choice of these methods of data collection and analysis -— 

Response:

According to the reviewer’s the choice of data collection and analysis has been justified. Please see Page 7 line 215-222; Page 14 line 408-417. 

Comment:

- finally, please read and discuss in your paper the newly published piece of Ariane Goetz et al on polycentricity governance and agroecology in the Mena region published in international journal of water resources development. I am sure this would be a good addition to your work.

Response:

The author is thankful to the reviewer for this comment. According to the reviewer’s suggestion, the citation has been included. Please see Page 3 line 99-101. 

Reviewer 2 Report

Comments and Suggestions for Authors

The manuscript provide interesting scientific information on the Mn application strategies to soybean. The referee thinks that this manuscript has a merit to be issued in Sustainability; however, the manuscript also has several deficiencies to be improved before acceptance for publication.

2.3 Determination of CL of Mn by graphical method

2.4 Determination of CL of Mn by statistical method

 Readers can not understand how you decided the CLs by those methods and “what the merits and demerits of those two methods without consulting the referenced papers. How could you estimate CL based on an averaging Bray’s % yield in the graphical method? What were population1 and population2 in the equation for R2? Add more detailed explanations so that the readers can understand the calculation methods without having to search the references.

Line 222: Correct “CSS1” to “CSS2” and “Population1” to “Population2”.

Line 229: The treatments were 50% CL (1.8 ug g-1 Mn), …

  How much amount of MnSO4 you mixed with soybean seeds? How much did those mixture rates equate to g per hectare?

Line 271: Considering the critical value of Mn in soils, …

  How much amount of Mn did “medium”, “optimum”, “high” and “very high” refer to? Which soil samples were grouped into medium, optimum, high, and very high?

Table 5.  A column expressing the soil sample names (“Sample” in Table 1~3, but “Soils” in Table 6) should be added.

Line 347: The critical level is a soil test level that indicates the division between responsive and non-responsive conditions to Mn application.

/Decided CLs of Mn in this study were 3.60 ug g-1 in the graphical method and 3.55 ug g-1 in the statistical method, respectively.

  Even in the soils that the initial Mn content was above 3.55 ug or 3.66 ug, for instance S14 and S15, application of Mn improved the grain yield as much as 15 % point. What conditions did the authors refer to as “responsive to Mn”?

Line 370: (Table 4) would be (Table 6).

Line 385: Ghasemian et al. showed that Mn addition …

  By the application of Mn, which yield components of soybean were improved in this study?

Author Response

Comment:

The manuscript provide interesting scientific information on the Mn application strategies to soybean. The referee thinks that this manuscript has a merit to be issued in Sustainability; however, the manuscript also has several deficiencies to be improved before acceptance for publication.

Response:

The authors are thankful to the reviewer for the comments.

Comment:

2.3 Determination of CL of Mn by graphical method

2.4 Determination of CL of Mn by statistical method

Readers can not understand how you decided the CLs by those methods and “what the merits and demerits of those two methods without consulting the referenced papers. How could you estimate CL based on an averaging Bray’s % yield in the graphical method? What were population1 and population2 in the equation for R2? Add more detailed explanations so that the readers can understand the calculation methods without having to search the references.

Response:

The authors have included referenced papers and discussed about the two methods. In fact, these

two methods are well established for determining critical limit of plant nutrients. The detailed explanations of the procedure for estimation of CL by both graphical and statistical methods are added in the section ‘Materials and Methods’. Please see Page 7 line 209-233; Page 8 line 234-252.

Comment:

Line 222: Correct “CSS1” to “CSS2” and “Population1” to “Population2”.

Response:

The authors have corrected these words. Please see Page 8 line 251.

Comment:

Line 229: The treatments were 50% CL (1.8 ug g-1 Mn), …

  How much amount of MnSO4 you mixed with soybean seeds? How much did those mixture rates equate to g per hectare?

Response:

In field trial, the treatments were 50% CL (1.8 µg Mn g-1 soil i.e. 3.6 kg Mn ha-1), 75% CL (2.7 µg Mn g-1 soil i.e. 5.4 kg Mn ha-1), 100% CL (3.6 µg Mn g-1 soil i.e. 7.2 kg Mn ha-1) and 125% CL (4.5 µg Mn g-1 soil i.e. 8.9 kg Mn ha-1). The required amount of MnSO4.H2O was 5.53 µg g-1 soil (i.e. 11.06 kg ha-1) for 50% CL, 8.29 µg g-1 soil (i.e. 16.58 kg ha-1) for 75% CL, 11.05 µg g-1 soil (i.e. 22.10 kg ha-1) for 100% CL, and 13.8 µg g-1 soil (i.e. 27.6 kg ha-1) for 125% CL. The authors have mentioned the amount of MnSO4 and the doses of Mn in the text. Please see Page 8 line 259-261 and 267-270.

Again, in pot trial, to add 0, 1, 2 and 4 µg Mn g-1 soil, the required amount of MnSO4.H2O was 0 µg g-1 soil (i.e. 0 kg ha-1), 3.07 µg g-1 soil (i.e. 6.14 kg ha-1), 6.14 µg g-1 soil (i.e. 12.28 kg ha-1), and 12.28 µg g-1 soil (i.e. 24.56 kg ha-1). Please see Page 7 line 197-199.

Comment:

Line 271: Considering the critical value of Mn in soils, …

How much amount of Mn did “medium”, “optimum”, “high” and “very high” refer to? Which soil samples were grouped into medium, optimum, high, and very high?

Response:

In the study, 5% of soils (S1) was grouped into medium (1.51-2.25 µg g-1 Mn), 25% (S2-S6) optimum (2.26-3.0 µg g-1 Mn), 20% (S7-S10) high (3.10-3.75 µg g-1 Mn) and 50% (S11-S20) very high classes (˃3.75 µg g-1 Mn). This is a conventional method of categorizing soils based on

nutrient level for fertilizer recommendation/application (Fertilizer Recommendation Guide,

2018). Please see Page 9 line 305-308.

Materials & Methods

Comment:

Table 5.  A column expressing the soil sample names (“Sample” in Table 1~3, but “Soils” in Table 6) should be added.

Response:

The word ‘Soil’ has been added in the column. Please see Page 4 line 152 (Table 1), Page 5 line 166 (Table 2); Page 6 line 183 (Table 3).

Comment:

Line 347: The critical level is a soil test level that indicates the division between responsive and non-responsive conditions to Mn application. /Decided CLs of Mn in this study were 3.60 ug g-1 in the graphical method and 3.55 ug g-1 in the statistical method, respectively.

  Even in the soils that the initial Mn content was above 3.55 ug or 3.66 ug, for instance S14 and S15, application of Mn improved the grain yield as much as 15 % point. What conditions did the authors refer to as “responsive to Mn”?

Response:

Soil is a complex and heterogeneous system. Plant responds to the added nutrient which is even

more easily available than extraction from the soil. The same we have been observing for some other nutrients like available P. In our study, we have mentioned the condition in the text. Please see Page 13 line 379-384.

Comment:

Line 370: (Table 4) would be (Table 6).

Response:

The authors have corrected the table number. Please see Page 14 line 405.

Comment:

Line 385: Ghasemian et al. showed that Mn addition …

  By the application of Mn, which yield components of soybean were improved in this study?

Response:

In this study, by Mn application, the number of pod and seed as well as seed weight was increased. Please see Page 14 line 431, 434-435.

Reviewer 3 Report

Comments and Suggestions for Authors

Please take into consideration the folowing recommendation:

            line 225-226: Please clarify the location of field trial A field trial conducted to assess soybean (BARI Soybean-6) response to Mn fertilization at AEZ 18 in Subornochar upazila (22°42' N, 91°05' E)” do you mean “Subornochar (22°42' N 91°01' E)” ?

            The fertilization which has done in the pot trial and field condition seems to be not exactly the same:

            lines 187-189 in pot trial: „According to FRG [33], N, P, K, S, Zn and B were applied from urea, triple super phosphate, muriate of potash, gypsum, zinc sulphate and boric acid, respectively at the dose of 27, 30, 40, 14, 1 and 0.5 kg 189 ha-1, respectively.”

            vs. in field condition: „27 kg ha-1 of N, 31 kg ha-1 of P, 53 kg ha-1 of K, 1 kg ha-1 of Zn 232 and 1.1 kg ha-1 of B were applied according to the soil test basis from urea, TSP, MoP, zinc sulphate and boric acid, respectively.”.  In „field conditions” sulfur lacks, probably due to the low pH? – but „in pots” the soil from the 20 locations has a variable pH – please see line 251 „Soil pH differed from 5.20 to 7.85” and gypsum was applied.

            in line 21 it is specified that „A pot experiment with 20 soils collected from three Agro-Ecological Zones (AEZs) of Bangladesh” but in line 185 „A number of 240 plastic pots with an eight-kilogram capacity (24 cm in diameter) were filled with soil” – it is not specified that soil from al the 20 locations was used

            If most of the sampled soil were classified as optimum to very high in Mn content (1.78 to 6.76 μg g-1), please see the stament in lines 268-272: ”The status of available Zn, Cu, Fe and Mn of the soils fluctuated from 0.32 to 1.70, 0.78 to 5.56, 30.6 to 135 and 1.78 to 6.76 μg g-1, respectively in studied soil set having mean values of 0.84, 2.12, 68.4 and 3.82 μg g-1, respectively. Zn level was very low to optimum. Considering the critical value of Mn in soils (1.0 μg g-1), 5% of soils was grouped into medium, 25% optimum, 20% high and 50% very high classes.” and please clarify why it is considered necessary to determine the critical limits for Mn as a target to improve the yield of soybean?

            It should be also taken into consideration that excessive content of certain nutrients can also block the uptake and utilization of other nutrients thus a balanced nutrient management should be analyzed and considered. It would be interesting to take into consideration the peak demand of Mn in soybean which, according to specialty literature, seems to be at the early seed filling stage.

            Lines 29-30 and lines 408-410: Please clarify and correlate the statements from the “Abstract” section: “The findings of the study could help predict soybean response to Mn fertilizer which is required for efficient fertilizer management practices.” with  the statements from the “Conclusion” section: “Nevertheless, it is tough to fix a constant value of critical limit of soil available Mn in various types of soils owing to soil heterogeneity, crop species and different varieties of a given species.

Author Response

Comment:

            line 225-226: Please clarify the location of field trial “A field trial conducted to assess soybean (BARI Soybean-6) response to Mn fertilization at AEZ 18 in Subornochar upazila (22°42' N, 91°05' E)” do you mean “Subornochar (22°42' N 91°01' E)” ?

Response:

The authors are sorry for the typing mistake. We have corrected the location of the field trial. Please see Page 8 line 256.

Comment:

            The fertilization which has done in the pot trial and field condition seems to be not exactly the same:

            lines 187-189 in pot trial: „According to FRG [33], N, P, K, S, Zn and B were applied from urea, triple super phosphate, muriate of potash, gypsum, zinc sulphate and boric acid, respectively at the dose of 27, 30, 40, 14, 1 and 0.5 kg 189 ha-1, respectively.”

  1. in field condition: „27 kg ha-1 of N, 31 kg ha-1 of P, 53 kg ha-1 of K, 1 kg ha-1 of Zn 232 and 1.1 kg ha-1 of B were applied according to the soil test basis from urea, TSP, MoP, zinc sulphate and boric acid, respectively.”.  In „field conditions” sulfur lacks, probably due to the low pH? – but „in pots” the soil from the 20 locations has a variable pH – please see line 251 „Soil pH differed from 5.20 to 7.85” and gypsum was applied.

Response:

Actually, the fertilizer doses for N, P, K, S, Zn and B (as basal), applied in the pot trial and field condition were the same. We have applied fertilizers in the pot based on the soil mass in one ha land. The corrected doses of fertilizers are mentioned in the text. Please see Page 8 line 262-265.

Comment:

            in line 21 it is specified that „A pot experiment with 20 soils collected from three Agro-Ecological Zones (AEZs) of Bangladesh” but in line 185 „A number of 240 plastic pots with an eight-kilogram capacity (24 cm in diameter) were filled with soil” – it is not specified that soil from all the 20 locations was used

Response:

The authors used 20 soils with 4 treatments and 3 replications; therefore, 20×4×3 = 240 plastic pots were used. The authors have mentioned and specified the soils in the text. Please see Page 7 line 187-192.

Comment:

            If most of the sampled soil were classified as optimum to very high in Mn content (1.78 to 6.76 μg g-1), please see the stament in lines 268-272: ”The status of available Zn, Cu, Fe and Mn of the soils fluctuated from 0.32 to 1.70, 0.78 to 5.56, 30.6 to 135 and 1.78 to 6.76 μg g-1, respectively in studied soil set having mean values of 0.84, 2.12, 68.4 and 3.82 μg g-1, respectively. Zn level was very low to optimum. Considering the critical value of Mn in soils (1.0 μg g-1), 5% of soils was grouped into medium, 25% optimum, 20% high and 50% very high classes.” and please clarify why it is considered necessary to determine the critical limits for Mn as a target to improve the yield of soybean?

Response:

Manganese is associated with numerous metabolic functions in plant cellular compartments and is of paramount importance as a micronutrient for plant growth. Soybean is a high Mn-responsive crop. In many regions of the world, soybeans are more often deficient in Mn than in other micronutrients, and respond well to Mn fertilizers when deficient. Diagnosis and prevention of any nutrient deficiency and toxicity require knowledge on critical value of that particular element in soil and plant. The critical limit can be used to find Mn-deficient soils from non-deficient and can provide information on the Mn status of soils. Therefore, to improve soybean yield, the critical limit of Mn should be considered. According to the comments of the reviewer, the authors have now clearly mentioned the reason in the text. Please see Page 14 line 418-430.

Comment:

            It should be also taken into consideration that excessive content of certain nutrients can also block the uptake and utilization of other nutrients thus a balanced nutrient management should be analyzed and considered. It would be interesting to take into consideration the peak demand of Mn in soybean which, according to specialty literature, seems to be at the early seed filling stage.

Response:

The authors have improved the text accordingly. Please see Page 14 line 448-453 and 428-430.

Comment:

            Lines 29-30 and lines 408-410: Please clarify and correlate the statements from the “Abstract” section: “The findings of the study could help predict soybean response to Mn fertilizer which is required for efficient fertilizer management practices.” with  the statements from the “Conclusion” section: “Nevertheless, it is tough to fix a constant value of critical limit of soil available Mn in various types of soils owing to soil heterogeneity, crop species and different varieties of a given species.”

Response:

The authors have revised the ‘Conclusion’ accordingly. Please see Page 15 line 466-471.

Round 2

Reviewer 1 Report

Comments and Suggestions for Authors

Improved